# The CXCR4-Dependent LASP1-Ago2 Interaction in Triple-Negative Breast Cancer

**DOI:** 10.3390/cancers12092455

**Published:** 2020-08-29

**Authors:** Augustus M. C. Tilley, Cory M. Howard, Sangita Sridharan, Boopathi Subramaniyan, Nicole R. Bearss, Sawsan Alkhalili, Dayanidhi Raman

**Affiliations:** Department of Cancer Biology, College of Medicine and Life Sciences, University of Toledo Health Science Campus, Toledo, OH 43614, USA; Augustus.Tilley@rockets.utoledo.edu (A.M.C.T.); cory.howard@rockets.utoledo.edu (C.M.H.); sangita.sridharan@rockets.utoledo.edu (S.S.); boopathi.subramaniyan@utoledo.edu (B.S.); nicole.bearss@utoledo.edu (N.R.B.); sawsalk@umich.edu (S.A.)

**Keywords:** Argonaute2, LASP1, CXCR4, Let-7a, triple negative breast cancer

## Abstract

**Simple Summary:**

CXCR4 is critically involved in triple-negative breast cancer metastasis. LASP1 was previously found to be necessary for CXCR4-dependent tumor cell invasion. Here we aimed to understand how LASP1 can have such a powerful role this process that is critical to metastasis. We found Ago2, a master regulator protein, to associate with LASP1 in response to CXCR4 activity. Furthermore, we found that this association was responsible for affecting the expression of proteins regulated by Ago2.

**Abstract:**

The CXCR4-LASP1 axis is an emerging target in the field of breast cancer metastasis. C-X-C chemokine receptor type 4 (CXCR4) mediates directed cell migration when activated by its cognate ligand CXCL12. LIM and SH3 Protein 1 (LASP1) is a critical node in the CXCR4 signaling pathway, as its deficiency blocks CXCR4-dependent Matrigel invasion. The mechanism by which LASP1 facilitates this invasive ability of tumor cells when CXCR4 is activated is unknown. Our previous proteomics work had revealed several components of the RNA interference (RNAi) machinery as being potential LASP1 interacting proteins. Here we report that argonaute 2 (Ago2), a protein with central involvement in RNAi, associates with LASP1 in triple-negative breast cancer (TNBC) cells. We demonstrate that LASP1 co-immunoprecipitates with Ago2 endogenously in a CXCL12-dependent manner, with further confirmation of this interaction by proximity ligation assay. Furthermore, this association is specific to CXCR4 as it can be abrogated by the CXCR4 antagonist, AMD3465. By GST-pulldown approach, we identify that LASP1 directly binds to Ago2 through its LIM and SH3 domains, and that this binding is dictated by the S146 and Y171 phosphorylation sites of LASP1. Additionally, the phosphorylation status of LASP1 affected tumor suppressor microRNA (miRNA) Let-7a-guided Ago2 activity. Levels of several endogenous targets of Let-7a were found to be altered including C-C chemokine receptor type 7 (CCR7), which is another critical chemokine receptor involved in metastasis to lymph nodes. Our results suggest a novel role for the LASP1-Ago2 module in shaping the RNAi landscape, functionally impacting the invasive ability of cancer cells.

## 1. Introduction

Triple-negative breast cancer (TNBC) has the lowest pathological complete response (pCR) of all breast cancers. Though it initially responds to therapy, it quickly becomes refractory to current neoadjuvant chemotherapies (NACT) involving platinum drugs, taxanes and anthracyclines [1,2,3]. TNBC is stratified into multiple subtypes, each having distinct molecular patterns [4,5,6]. One of the defining features of TNBC is its high incidence of metastasis [7], particularly to the lungs, bone and brain when compared with other types of breast cancer [3]. Primary, localized breast cancer is rarely lethal, but it can metastasize to other critical organs throughout the body. Between metastasis and chemotherapy resistance [8], treating TNBC patients is an extreme challenge with a generally poor clinical outcome. Therefore, it is imperative to identify novel therapeutic targets critically involved in the metastatic process, in the hopes of developing drugs effective at improving the quality of life and longevity of TNBC patients.

LIM and SH3 Protein 1 (LASP1) is an adaptor protein that is overexpressed in breast cancer [9], and plays a critical role in tumor cell migration [10] and invasion [11]. LASP1 was identified as an adaptor protein as it facilitates protein-protein interactions and interacts with a significant number of proteins [12]. The effect of LASP1 on focal adhesion dynamics [13] through its interaction with zona occludens 2 (ZO-2) [14] and zyxin [15], as well as its impact on motility through association with the chemokine receptor CXCR4 [16] and vimentin [17], suggests an intricate role for LASP1 in breast cancer metastasis. In addition to breast cancer, LASP1 has also been shown to be important in the migration of lung and prostate cancer cells [12]. This role of LASP1 in cancer suggests a critical role in cancer progression and metastasis.

LASP1 is comprised of four domains: an amino-terminal LIM domain followed by two nebulin repeats (NR), and a linker domain (LD) connecting the NRs to the carboxyl-terminal *src* homology 3 (SH3) domain [18,19,20]. The internal NR repeats are responsible for binding to filamentous actin (F-actin) [19], while the LIM and SH3 domains interact with multiple proteins. The *Lin*, *Isl* and *Me*c [21] (LIM) domain mediates the direct binding to chemokine receptors CXCR1-4 [16], while the SH3 [22] domain was responsible for binding to vimentin [17], zyxin [15] and ZO-2 [14]. Most of the known LASP1 interacting proteins are either directly or indirectly involved in the process of cell migration, and their binding dynamics seem to be dependent on the phosphorylation status of LASP1. Of the four domains of LASP1, only the LD contains validated phosphorylation sites (S146 and Y171). S146 is phosphorylated by protein kinases A and G (PKA and PKG), dephosphorylated by protein phosphatase 2B (PP2B) [14], and is involved in re-localization of LASP1 from cell membrane extensions and cell-cell contacts to the cytosol [23]. The non-receptor tyrosine kinases c-*Src* [24] and c-*Abl* [13,25] phosphorylate LASP1 on Y171 and prevent its re-localization from the leading edge to focal adhesions [13]. These sites were found to be responsible for the LASP1-CXCR4 interaction, where S146 phosphorylation of LASP1 was indicative of binding to CXCR4 and Y171 phosphorylation was responsible for dissociation from CXCR4 [26].

CXCR4 is a chemokine receptor involved in several processes during embryogenesis, brain development [27], early stem cell migration [28], and maintenance of hematopoietic stem cells in postnatal life [29]. The role of CXCR4 in each of these processes is related to its underlying function of directional migration along the CXCL12 chemokine gradient [30]. In addition to its role in normal physiology, CXCR4 also plays an important role in disease progression and metastasis of several different types of solid tumors including breast cancer [31,32,33]. In breast cancer, CXCR4 is overexpressed and is responsible for metastasis to predilection sites in the body that are enriched for CXCL12 including the lungs, bone marrow and brain [34,35].

When LASP1 was stably knocked down in MDA-MB-231 (MDA-MB-231: ATCC^®^ HTB-26™, Manassas, VA, USA) breast cancer cells, there was an ablation of CXCL12-dependent invasion through Matrigel^®^ (Matrigel: Discovery Labware, Inc., Bedford, MA, USA) [11]. This profound impact suggests that LASP1 has a critical role in facilitating the necessary downstream activity required to allow CXCR4 to drive invasion. As LASP1 is an adaptor protein it is likely that its loss prevents critical protein-protein interactions downstream in the CXCR4 signaling pathway, leading to the loss of the ability of tumor cells to invade through Matrigel^®^.

Duvall-Noelle et al. [11] performed a Multidimensional Protein Identification Technology (MudPIT) proteomics analysis to identify novel proteins that associate with LASP1 in response to stimulation of TNBC cells with CXCL12. The dataset identified several proteins involved in RNAi as potential LASP1 interacting proteins, which are recruited by the RNA-induced silencing complex (RISC) to aid in degradation or repression of messenger RNAs (mRNAs). The minimal functional form of the RISC is an Argonaute protein bound with a miRNA [36,37,38]. X-linked ribonuclease 1/2 (XRN1/2), decapping protein 1 (DCP1), and GW182 / trinucleotide repeat-containing adaptor 6A (TNRC6A) are recruited after an Argonaute protein identifies a target RNA with sufficient sequence complementarity to the miRNA guide strand [39,40,41]. GW182 directly binds the Argonaute proteins and facilitates recruitment of additional proteins involved in degradation or repression of the RNA target [42], including the deadenylase complexes PAN2-PAN3 and CCR4-NOT1 [43] as well as the polyA binding protein PABPC1 [44]. The Argonaute protein family is comprised of two subfamilies; the Ago subfamily containing Argonautes 1–4 (Ago1–4), and the Piwi subfamily containing HIWI1-3 and HILI [45,46]. While all members of the Argonaute protein family utilize small RNAs to perform RNAi, only Ago2 has catalytic activity. Ago2 exhibits endonuclease activity when the guide RNA has proper sequence complementarity with the target RNA. This allows Ago2 to directly cleave mRNA targets without recruitment of other proteins involved in the RNAi process [45,46].

In this study, we explore the possibility that LASP1 interacts with Ago2 either directly or indirectly through the other RNAi associated proteins identified in the MudPIT analysis. We demonstrate through various approaches that LASP1 interacts with Ago2 which can be stimulated by CXCR4 activity, and that this interaction is caused by a direct binding between the two proteins. Furthermore, we reveal that this interaction preferentially occurs when LASP1 is dephosphorylated at the S146 site and phosphorylated at the Y171 site. We also found that this interaction impacts pro-motility and cell cycle proteins that are targets of Ago2-based RNAi regulation including the CC chemokine receptor CCR7, cyclin D1 and eukaryotic translation initiation factor G2 (eIF4G2). Based on these results and previous findings we propose a model where CXCR4 signaling activity stimulates binding of LASP1 to Ago2, which promotes cell motility by shifting RNAi-based repression away from pro-motility targets.

## 2. Results

### 2.1. LASP1 Associates with Ago2 through Its LIM and SH3 Domains

To determine if LASP1 associates with Ago2, we created a series of LASP1 constructs that would allow us to probe the interaction. These constructs included the full length LASP1 protein, its LIM domain, the combination of its NR and linker domain (LD and NRLD for the combination), and the SH3 domain (Figure 1A). The constructs were GST-tagged, expressed, and purified from bacteria using glutathione-agarose affinity chromatography. The purity of GST-LASP1 and its domains were verified by SDS-PAGE followed by Imperial Blue protein stain (Figure 1B). To probe the LASP1-Ago2 interaction we employed a GST-pulldown approach, where we incubated the GST-LASP1 and its domains with total protein lysate from 231S cells. 231S is a variant of the MDA-MB-231 (MDA-MB-231: ATCC^®^ HTB-26™, Manassas, VA, USA) cell line isolated by flow sort for high cell surface expression of CXCR4 [11]. Immunoblotting of the pulldown revealed that endogenous Ago2 associates with full length LASP1 as well as its LIM and SH3 domains (Figure 1C). We also found that Ago1 associated with LASP1, although it only interacted through the SH3 domain (Appendix A). This revealed that more members of the Ago family interact with LASP1 and that the LASP1-Ago2 interaction may be unique among them, as Ago2 is the only family member with catalytic activity.

### 2.2. LASP1 Directly Binds to Ago2 through Its LIM and SH3 Domains

In order to test the possibility of a direct binding interaction between LASP1 and Ago2, we incubated purified GST-LASP1 and its domains with purified recombinant Ago2. We found that Ago2 was capable of directly binding to LASP1 in a concentration dependent manner (Figure 1D). We also tested to assess the specific domains that were involved in the binding interaction. We show that both the LIM and SH3 domains were capable of directly binding to Ago2 (Figure 1E). This mirrored the result that we found when using total protein lysates, suggesting that any endogenous interaction would be a result of a direct binding between the two proteins.

### 2.3. LASP1 Endogenously Interacts with Ago2 in a CXCL12-Dependent Manner

To determine if the interaction between LASP1 and Ago2 occurs endogenously, we performed co-immunoprecipitation (Co-IP) and proximity ligation assays (PLA). For the co-immunoprecipitation we stimulated the 231S cells with 20 nM CXCL12 over the course of 60 min and found two interaction peaks, with the first peak occurring at 30 min and the second peak at 50–60 min (Figure 2A). To validate that this effect was downstream of CXCR4 activity, we utilized the CXCR4-specific allosteric inhibitor AMD3465. AMD3465 is a potent inhibitor with high specificity for CXCR4 [47,48,49], which has been shown to strongly inhibit CXCR4 function in breast cancer [50]. We found that the addition of AMD3465 prior to stimulation with CXCL12 ablated the peak observed at 30 min. This suggested that the interaction seen at 30 min was specific to CXCR4.

Next, we employed PLA to examine the association between LASP1 and Ago2 following stimulation with CXCL12 in situ. To this end we utilized a different variant of the MDA-MB-231 (MDA-MB-231: ATCC^®^ HTB-26™, Manassas, VA, USA) cell line termed MDA-Bone-Un, similarly isolated for its high cell surface expression of CXCR4 [11]. Cells were stimulated with 20 nM CXCL12 for 30 min, either in the presence or absence of AMD3465. We observed a dramatic increase in both the number and intensity of PLA interactions (dots) when stimulated with CXCL12 (Figure 2B). We also found that the amount of interactions could be diminished when AMD3465 was added prior to stimulation. We quantified the number of interactions per cell for each condition and found a 2.5 to 3-fold increase in the +CXCL12 condition over -CXCL12. The CXCL12-dependent increase in PLA interactions was abrogated when AMD3465 was added (Figure 2C). The PLA and co-immunoprecipitation results agreed, revealing a specific LASP1-Ago2 interaction in situ that can be facilitated in the presence of CXCL12. Interestingly, the PLA also revealed that this interaction occurs throughout the cell including the nucleus (Appendix A). This suggests a global effect of the CXCR4-LASP1 axis on Ago2 and RNAi as Ago2 has different functions in the nucleus, implying that LASP1 may affect more than one function of Ago2.

### 2.4. The Phosphorylation Status of LASP1 Dictates the LASP1-Ago2 Interaction

Protein-protein interaction dynamics can be influenced by the phosphorylation status of the proteins involved in the interaction. The changes in the phosphorylation status of LASP1 in response to CXCR4 activity [26] may be a driving factor in the LASP1-Ago2 interaction. To investigate this possibility, we created four phosphorylation mutants of LASP1, one phospho-null for each site (S146A and Y171F) as well as one phospho-mimetic for each site (S146D and Y171D) (Figure 3A). These GST-tagged phospho-mutant LASP1 proteins (Figure 3B) were employed in pulldown assays. Proteins with LIM domains are known to dimerize through the domain’s zinc-finger motif [51,52,53], and while there has currently been no reported LASP1 homodimerization [54,55] the possibility remains. If LASP1 were to homodimerize, there is a strong possibility that it could homodimerize two different phosphorylation states of LASP1. This possibility would affect any pulldown experiment for Ago2 that included endogenous LASP1. In order to eliminate this possibility, we decided to employ a cell line free of endogenous LASP1 for this assay. In our previous work we published a LASP1 knock-out (KO) in HEK 293 cells [56]. Using the same technique, we employed total cell lysates from previously generated LASP1 KO in Bone-Un cells [57] which we incubated with our GST-tagged WT and mutant LASP1 proteins. We found that Ago2 preferred S146A over S146D as well as Y171D over Y171F (Figure 3C). Previous work showed that the phosphorylation at Y171 increases and S146 phosphorylation decreases with CXCR4 stimulation [26], therefore this result is consistent based on our finding that the LASP1-Ago2 interaction increases in response to CXCR4 stimulation. To validate this result, we repeated this phospho-mutant pulldown with purified recombinant Ago2 instead of protein lysates and found the same pattern of association (Figure 3D). We also tested 231S protein lysates to see if the endogenous LASP1 would interfere with the result, and we found that Ago2 preferentially associates with S146A over S146D as well as preferring Y171F over Y171D, with Y171F being the strongest associator (Appendix A). This difference in result suggests that the endogenous LASP1 does alter the association profile, hinting at the possibility of endogenous WT LASP1 homodimerizing with GST-LASP1 WT or its mutants.

Next, we wanted to determine if creating a double-phosphorylation mutant could improve the binding affinity of Ago2 to LASP1. We engineered four different combinations of double-mutants termed SAYF (S146A + Y171F), SAYD (S146A + Y171D), SDYF (S146D + Y171F), and SDYD (S146D + Y171D) via site-directed mutagenesis (Figure 3A). These double mutants were also GST-tagged, and then purified using glutathione agarose beads (Figure 3E). We performed the GST-pulldown analysis by incubating the double mutants with protein lysate from our Bone-Un LASP1 knockout cells. As with the single mutant knockout pulldown we observed the expected association pattern with SAYD being the strongest associator and SDYF being the weakest (Figure 3F). We also performed a direct binding pulldown to validate the Bone-Un knockout lysate experiment and found the expected result where SAYD was the strongest binding partner and SDYF the weakest (Figure 3G). We repeated this experiment with 231S lysates to see if endogenous LASP1 would alter the binding pattern just as it did in the single mutant pulldown experiment. What we found was this pulldown gave a similar result to that of the single-mutant pulldown, where the SAYF double-mutant had the strongest association and the SDYF mutant had the weakest (Appendix A).

### 2.5. Let-7a and MiR-100 Targets Are Altered in TNBC

The work of Bridge et al. [58] revealed a unique connection between the LIM domain containing protein LIMD1 and the Argonaute family members, where LIMD1 facilitated the interaction between GW182 and Ago2. Their work also revealed that the Let-7a and miR-100 microRNAs (miRNAs) had their functionality altered in response to loss of LIMD1. These two miRNAs are both involved in cancer progression and are utilized by Ago2 [58]. Let-7a is a tumor suppressor that is downregulated in breast cancer progression [59,60,61]. MiR-100 is interesting in that it is involved in the epithelial-mesenchymal transition [62] but also suppresses breast cancer progression [62,63,64]. To better understand the LASP1-Ago2 interaction we wanted to identify motility-related protein targets of Ago2 that are upregulated in breast cancer. To identify such targets, we first picked predicted targets from miRDB (http://www.mirdb.org/) [65,66] and then performed an analysis using the Oncomine™ (https://www.oncomine.org/) [67,68] software to assess their differential gene expression. In this analysis, both “The Cancer Genome Atlas” [69] and Curtis data sets [70] were employed. We compared the expression of the targets in both invasive lobular and invasive ductal carcinomas (ILC and IDC respectively) to normal breast tissue for both Let-7a and miR-100 targets. We evaluated the following targets for Let-7a: CCR7, a chemokine receptor and validated Let-7a target involved in breast cancer metastasis to lymph nodes [59,71,72]; eukaryotic translation initiation factor 4G2 (eIF4G2), a predicted target involved in cancer progression and metastasis [73,74]; cyclin D1, a validated target critically involved in progression, chemoresistance and survival [75,76,77]; and vinculin a predicted target involved in directionality of migration and focal adhesion dynamics [78,79]. Target selection for miR-100 was more challenging as there are far less validated targets and even less related to cancer progression, thus we analyzed the validated survival related target mammalian target of rapamycin (mTOR) [80,81], as well as Ago1 and Ago2 which are predicted targets of miR-100. We found that all four Let-7a targets were upregulated in both cancer types for both datasets (Figure 4A–D). For the miR-100 targets there was far less of a trend with only Ago2 being upregulated in three of the four datasets. Ago2 was found to be upregulated in the Curtis Breast Invasive Ductal Carcinoma (IDC) dataset (Figure 4A) as well as both of the “The Cancer Genome Atlas” (TCGA) datasets (Figure 4C,D), but no significant change was noted in the Curtis Breast Invasive Lobular Carcinoma (ILC) dataset (Figure 4B). For both mTOR and Ago1 no significant difference was measured in both Curtis datasets and the TCGA ILC dataset (Figure 4A,B,D), but there was an identified significant difference in the TCGA IDC dataset (Figure 4C). This suggests that Let-7a targets will be strong downstream targets of the LASP1-Ago2 interaction as they are upregulated in advanced breast cancer, and miR-100 targets outside of Ago2 itself may not be strong targets.

### 2.6. Let-7a Based Target Repression is Altered in a LASP1-Dependent Manner

To determine if LASP1 has an impact on Ago2 functionality, we designed a luciferase reporter assay utilizing our phospho-mutant LASP1 proteins. We shuttled our LASP1 double phospho-mutant constructs (SAYD: dominant active and SDYF: dominant negative regarding Ago2 binding) into the mammalian expression vector pcDNA3.0 and electroporated them into Bone-Un LASP1-KO cells. Four different experimental stable Bone-Un cell lines were created including WT-LASP1 rescue, SAYD dominant-active, SDYF dominant-negative, and empty pcDNA 3.0 vector control (EV). Each of these four cell lines (Figure 5A) was transiently transfected with five previously published [58] dual-luciferase expressing vectors including empty vector Renilla luciferase (EV Luc), Let-7a, miR-100 targeted (miR-100 Tar), miR-100 not-targeted (miR-100 NT), and mTOR. EV Luc lacked a 3′-UTR while Let-7a, miR-100 Tar, and miR-100 NT all had artificial 3′-UTRs consisting of multiple corresponding miRNA target sites [58]. The mTOR vector contained the full 3′-UTR of the mTOR mRNA to compare with the miR-100 vectors. These vectors also contained the firefly luciferase gene under a different promoter within the same plasmid. The expression of firefly luciferase as a fusion protein with Renilla luciferase was employed as the normalizing internal control. The Let-7a construct results were compared against the EV Luc control (Figure 5B) and revealed 2-fold or greater increases in all the experimental conditions over the WT rescue cell line. Interestingly, both the EV cell line and the dominant-negative SDYF cell lines both had increased Renilla luciferase expression over the WT rescue cell line. This was contrary to expectation as we expected that decreased LASP1-Ago2 interaction would lead to higher activity of Let-7a and therefore lower expression of migration and metastasis related targets. The miR-100 Tar and mTOR luciferase readings (Figure 5C,D) were compared against the miR-100 NT and showed no robust change between different cell lines. This suggested that LASP1 affects the ability of Ago2 to perform RNAi, and that this effect could be specific to motility-regulating miRNAs.

### 2.7. The LASP1-Ago2 Interaction Impacts Expression of Let-7a Targets

To discern if the LASP1-Ago2 interaction has an impact on translation of let-7a targets we utilized our model LASP1 double mutant rescue cell lines to test for endogenous Let-7a targets. Lysates were probed for different Let-7a targets including eIF4G2 (Figure 6A), vinculin, CCR7, and cyclin D1 (Figure 6B) in addition to LASP1, Ago2 and β-tubulin loading control. Each of the four targets revealed a similar trend where introduction of the dominant active SAYD had stronger expression of the targets as compared to the WT rescue cell line, and the dominant-negative SDYF cell line had moderate to highly decreased levels compared to the WT rescue. The empty vector control was variable across all four targets with expression being lower, similar and higher than the WT. We also confirmed that Ago2 levels did not change between cell lines, confirming our luciferase result that miR-100 targets were not altered (Figure 6A). These results were in line with what we expected, where the dominant active LASP1 favored an increase LASP1-Ago2 interaction which promoted expression of prometastatic targets, while dominant negative decreased the LASP1-Ago2 interaction and allows Ago2 to perform normal repression of prometastatic targets.

## 3. Discussion

In this study, we identified a novel interaction between LASP1 and Ago2 in TNBC cell lines in vitro, whose association could be stimulated through CXCR4 activity. Furthermore, we demonstrate that this interaction occurs in a phosphorylation-dependent manner. This may have a powerful impact upon Let-7a-based RNAi. Taken together, our results suggest that CXCR4 utilizes LASP1 to prevent Let-7a based mRNA repression leading to an increase in expression of cell motility and metastasis related targets. A proposed model behind this signaling pathway is provided (Figure 6C). Metastasis is a highly complex process with chemokines and their receptors at the heart. While chemokines and their receptors have been studied extensively, their complex signaling pathways still leave much to be discovered. For the first time, we connect chemokine receptor CXCR4 signaling to RNAi and implicate this connection in the dramatic changes that occur during the migration and invasion processes of cells.

Duvall-Noelle et al. [11] identified multiple components of the RISC as possible LASP1 interactors. While we only tested and found Argonautes 1 and 2 as LASP1 associating proteins (Figure 1C–E and Appendix A), it is likely that more of those components interact. Bridge et al. [58] found that the interaction of GW182 with Ago2 was affected by the LIM domain containing protein LIMD1. That finding in combination with the proteomics analysis strongly suggests that LASP1 will also bind to GW182. This has interesting implications for LASP1 acting as a binding mediator between Ago2 and GW182 it could have a strong impact on modulating RISC-dependent RNAi in response to a chemoattractant like CXCL12. Furthermore, if LASP1 were to be validated as a binding partner for more of the RISC components it would suggest the possibility that LASP1 may be capable of altering RISC formation. This possibility also prompts the question: are there more LIM proteins that interact with the RISC, and if so, do different LIM protein interactions with RISC components lead to differential repression or degradation of target RNAs?

Our results link the LASP1-Ago2 interaction to chemokine receptor signaling. We found that CXCR4-CXCL12 signaling stimulated the LASP1-Ago2 interaction (Figure 2), indicating a possible functional relationship between the two proteins in the context of chemotactic cell migration. The changes in phosphorylation status of LASP1 that occur in response to CXCR4 activation may transduce the signaling from active CXCR4. Thus, we tested and found that the phosphorylation status of LASP1 affected its ability to bind to Ago2 (Figure 3). We identified that Ago2 preferentially binds to the SAYD phospho-mutant, while it has the least association with the SDYF phosphorylation mutant. Based on previous findings this matches what happens to LASP1 before and after CXCR4 activation. Before activation LASP1 is strongly phosphorylated on S146 and not phosphorylated on Y171, while the opposite pattern is observed after activation of CXCR4.

While our results with our LASP1 phospho-mutants indicated the possibility of a functional purpose to the interaction there was still the possibility that this was a passenger interaction caused by the change in phosphorylation status. To test this possibility, we created LASP1 phospho-mutant expressing cell lines on a LASP1-KO background to probe if there was a function to the LASP1-Ago2 interaction (Figure 5A). We also found that the targets of tumor suppressor Let-7a had a higher expression in TNBC than in normal breast tissue suggesting decreased Let-7a activity (Figure 4). Based on the findings with LIMD1 there was a decent possibility that LASP1 would have a similar effect, so we decided to investigate this possibility by analyzing reporter expression in response to expression of mutant LASP1. We used a luciferase reporter assay with different artificial 3′-UTR target sites as well as an mTOR 3′-UTR and found that by mutating the phosphorylation status of LASP1 we were able to increase the expression of Let-7a targets (Figure 5B). This suggests that LASP1 inhibits Let-7a’s tumor suppressor function in response to CXCR4 activity in order to allow for induction of proteins that are involved in the process of cell motility and invasion. Interestingly, this phenomenon was only seen with Let-7a and not with miR-100 (Figure 5D), suggesting that this phenomenon may occur to specific miRNA interactions with Ago2 that was modulated by LASP1 instead of a global effect with all LIM domain containing proteins. Furthermore, the unchanged protein level of Ago2 supports that our Let-7a findings are specific, as Ago2 is a target of miR-100. From a broader perspective, our results imply that CXCR4 signaling may utilize RNAi to help induce the dramatic change in cell phenotype that is seen during cell migration.

The profile of the endogenous targets Let-7a (Figure 6A,B) differed from the results of our luciferase reporter assay. In the Western blots the dominant-negative SDYF had a significant loss of Let-7a target expression compared to the WT rescue, which is the result we had expected to observe in the luciferase assay. This difference in results can be explained by the differences between our artificial 3′-UTR with multiple tandem let-7a binding sites in the luciferase reporter construct compared to a natural 3′-UTR. In our results the artificial miR-100 targeted 3′-UTR (Figure 5C) had a significantly different set of values for the luciferase readout compared to the reporter with the added mTOR 3′-UTR (Figure 5D). While there was no significant change between cell lines in either condition there was a dramatic change in luciferase expression if you compare each condition’s relative luciferase levels (mTOR vs. miR-100 targeted). In a natural 3′-UTR it is possible to have target sites for multiple miRNAs, and the bases succeeding the miRNA seed sequence have a powerful effect on what ultimately happens to the target RNA [82]. In our reporter assay, our artificial Let-7a 3′-UTR had several tandem Let-7a target sites with near complete sequence complementarity to the actual Let-7a miRNA. Our construct’s seed sequence has a single nucleotide deletion compared to what the normal let-7a target site would be, as defined by the miRbase website (http://www.mirbase.org/). The work of Becker et al. [82] found that determination of whether the RNA target is repressed/degraded by the RISC or cleaved by Ago2 itself was determined by matching of bases past the seed sequence. Therefore, the difference between our luciferase assay and the endogenous biological target immunoblot profile is likely due to differences in the contextual information surrounding the Let-7a seed sequence between our artificial reporter and a natural 3′-UTR of the RNA target.

The expression of the endogenous Let-7a targets modulated by the LASP1 phospho-mutants have important functions in many steps of breast cancer progression and metastasis. The chemokine receptor CCR7 is a key protein involved in the metastasis of tumor cells to sentinel lymph nodes where there is an abundance of its ligand CCL21. This is the first report connecting the activity of the chemokine receptor CXCR4 to the expression of another chemokine receptor. Both CXCL12-CXCR4 and CCL21-CCR7 axes are majorly involved in metastasis of a variety of solid tumors. Cyclin D1 is involved in tumor cell proliferation, survival of the metastatic seed, tumor expansion and drug resistance. eIF4G2 is involved in drug resistance especially in non-small cell lung carcinoma cells [83] as well as metastasis and invasion in osteosarcoma cells. Vinculin is involved in focal adhesion dynamics and actin bundling, which regulates cell migration [84]. Overall, the Let-7a targets impacted by the LASP1-Ago2 interaction are involved in cancer progression, therapy resistance and metastasis.

With this body of work, we have established that LASP1 is a binding partner of Ago2 that has functional relevance to TNBC cell motility in response to CXCR4 activity, but our results only look at a very narrow scope of what could be affected by the LASP1-Ago2 interaction. In addition to the possibility that LASP1 has an impact upon RISC formation, it is also probable that the LASP1-Ago2 interaction also impacts the function of other miRNAs as well. In order to directly study what the interaction may affect it will be critical to understand mechanistically how LASP1 affects the function of Ago2. Ago2 is a master regulator of protein translation through miRNA-guided RNAi, but Ago2 is a very complex protein with effects in every step of protein biogenesis. Ago2’s first effects on protein biogenesis is in the nucleus [85,86] where it plays a role in transcription through chromatin alteration [87,88] and differential splicing [89]. Outside of its direct impact on protein biogenesis, Ago2 can also indirectly impact the it by having an effect on miRNA biogenesis by cleaving premature miRNA into their mature form [90,91]. Each of these possibilities will have to be explored before we can pinpoint which functions of Ago2 that LASP1 has an impact on as we saw PLA interactions in many subcellular compartments including the nuclei in TNBC cells (Figure 2B and Appendix A).

The CXCR4-LASP1 axis is an emerging pathway related to functions critical for breast cancer metastasis. Here, we connected this high impact pathway to Ago2, one of the central regulators of protein biogenesis. Through further investigation of the mechanism behind how LASP1 affects Ago2 functionality, we hope that we can either disrupt this pro-tumorigenic interaction or identify other novel therapeutic targets along this pathway. In summary, we report a novel interaction for LASP1 with Ago2 and this can be modulated by the activity of CXCR4. Additionally, we demonstrate that the expression of endogenous Let-7a targets can be modulated by the activity of the CXCR4-LASP1 axis.

## 4. Materials and Methods 

### 4.1. Cell Culture

All cell lines generated and utilized were derived from the parental MDA-MB-231 human breast cancer cell line (MDA-MB-231: ATCC^®^ HTB-26™, Manassas, VA, USA). The MDA-MB-231S (231S) variant was isolated by fluorescence-activated cell sorting (FACS) for high cell surface expression of CXCR4, while the Bone-Un variant was isolated from a metastatic bone lesion in a mouse xenograft model and validated as expressing high cell surface CXCR4 expression by FACS Both variant cell lines are described in previous work [11]. The LASP1 Bone-Un knockout cell line isolation and cultivation was performed in the same manner [57] as the HEK293 cell LASP1 knockout that was done and described in a previous work [56]. All cell lines including the LASP1 phospho-mutant rescues were cultured in Dulbecco’s modified eagle medium (DMEM) supplemented with 4 mM *L*-glutamine, + 4,500 mg/L glucose, and sodium pyruvate (GE Healthcare Life Sciences, Pittsburgh, PA, USA, Cat# SH30243.01), with added 10% heat-inactivated fetal bovine serum (FBS) (Denville Scientific, Swedesboro, NJ, USA, Cat# FB5001-H) and Penicillin (100 I.U.)/Streptomycin (100 μg/mL) (Corning, Corning, NY, USA, Cat# 30-002-CI).

### 4.2. Engineering of GST-LASP1 Domain Constructs

The full length human LASP1 and each of its domain constructs (LIM, NRLD, and SH3) were inserted into the pGEX-6P-1 bacterial expression vector (GE Healthcare Life Sciences, Pittsburgh, PA, USA, Cat# 28954648) using the BamHI and XhoI cloning sites. Primers for each respective construct are as follows: for the full length LASP1 construct (amino acids 1–261) 5′-CTAGCTGGATCCATGAA CCCCAACTGCGCC-3′ (forward) and 5′-CTAGCTCTCGAGTCAGATGGCCTCCACGTA-3′ (reverse); for the LIM domain construct (amino acids 1–60) 5′-CTAGCTGGATCCATGAACCC CAACTGC-3′ (forward) and 5′-CTAGCTCTCGAGTCACTGCTTGGGGTAGTG-3′ (reverse); for the NRLD domain construct (amino acids 52–199) 5′-CTAGCTGGATCCTGCAACGCACACTAC-3′ (forward) and 5′-CTAGCTCTCGAGTCAGGCGCTGCGCTGTATG-3′ (reverse); and for the SH3 domain construct (amino acids 200–261) 5′-CTAGCTGGATCCGGTGGTGGCGGGAAG-3′ (forward) and 5′-CTAGCTCTCGAGTCAGATGGCCTCCACGTA-3′ (reverse). Each construct was amplified by polymerase chain reaction (PCR) prior to insertion into the pGEX-6P-1 vector. Ligated vector-construct verity was performed by commercial sequencing (Eurofins Genomics, Louisville, KY, USA). All four vector-constructs and the empty vector (EV) control were introduced into the Rosetta variant of the E.Coli BL21 strain for production, with purification done by using glutathione-agarose beads (Thermo Scientific, Rockford, IL, USA, Cat# 16100), with the previously described protocol [92]. Verification of the purified protein product was performed by eluting and analyzing glutathione-agarose bead-bound GST proteins by 10% SDS-PAGE, followed by staining for protein with Imperial™ Protein Stain (Thermo Scientific, Rockford, IL, Cat# 24615). 

### 4.3. GST-Pulldown/Direct Binding Assays

Total cell lysates for the GST-pulldown assays were extracted from 231S cells by lysing on ice using either lysis buffer (100 mM NaCl, 50 mM Tris pH 8.0, 0.1% Deoxycholate, 0.1% IGEPAL CA-630 and 5 mM EDTA) or Co-IP buffer (20 mM HEPES pH 7.9, 10 mM NaCl, 1 mM MgCl2, 0.2 mM EDTA, 0.35% Triton X-100) supplemented with protease inhibitor cocktail (Sigma-Aldrich, St. Louis, MO, USA, Cat# P8340-5ML), phosphatase inhibitor cocktail 2 (Sigma-Aldrich, St. Louis, MO, USA, Cat# P5726), and phosphatase inhibitor cocktail 3 (Sigma-Aldrich, St. Louis, MO, USA, Cat# P0044). Total protein lysates were employed for the LASP1 domain pulldowns and direct binding assays, while the Co-IP buffer lysates were used in the phospho-mutant pulldowns/direct binding assays. For the phospho-mutant pulldowns with LASP1 knockout lysates, Bone-Un LASP1 knockout cells were lysed in co-immunoprecipitation buffer and total protein lysates were collected. Total protein in the lysates was quantified by Bradford assay (Bio-Rad, Hercules, CA, USA, Cat# 5000006). For the both the pulldown and direct binding assays, 1.5 nmol of each GST-LASP1 fusion protein bound to glutathione-agarose beads was resuspended in lysis buffer and incubated with either 1 mg of total protein lysate or 100–200 ng of purified recombinant Ago2 respectively (Active Motif, Carlsbad, CA, USA, Cat# 31486) for 2 h at 4 °C. Beads were subsequently washed with lysis buffer, with bound proteins being eluted and then analyzed by 10% SDS-PAGE followed by Western blotting.

### 4.4. Co-Immunoprecipitation Assay

Lysates were collected from 231S cells that were serum starved for 1 h and stimulated with 20 nM CXCL12 (PeproTech, Rocky Hill, NJ, USA, Cat# 300-28A) over 60 min. For the AMD3465 (CXCR4 antagonist, Sigma-Aldrich, St. Louis, MO, USA, Cat# SML1433-5MG) condition, a final concentration of 100 nM AMD3465 was added to the serum free media after 30 min of serum starvation. At time of collection, cells were washed in ice-cold phosphate-buffered saline, pH 7.5 (PBS) and total cell lysates were collected on ice in Co-IP buffer supplemented with protease and phosphatase inhibitor cocktails. The total protein was quantified by Bradford assay. 1 mg of total lysate was used per reaction condition. To the lysates, 2 µg of mouse anti-LASP1 antibody (Biolegend, San Diego, CA, USA, Cat# 909301) and 20 µL of protein G Dynabead™ in a 50% slurry (Invitrogen, Carlsbad, CA, USA, Cat# 10003D) were added, with the final volume of the immunoprecipitation being brought to 1 mL. 2 µg of monoclonal mouse IgG1 (Cell Signaling Technology, Danvers, MA, USA, Cat# 5415S) was used as an isotype control for the mock immunoprecipitation. After mixing lysates, antibodies, and beads together, the immunoprecipitation reaction proceeded for 2 h at 4 °C. After incubation, beads were washed in Co-IP buffer, with bound proteins being analyzed by 10% SDS-PAGE followed by immunoblotting.

### 4.5. Proximity Ligation Assay

The PLA assay was performed according to the manufacturer’s recommendations that accompanied the Duolink™ In Situ red fluorescent kit (Sigma-Aldrich, St. Louis, MO, USA, Cat# DUO92101-1KT). This Duolink™ kit was used for all PLAs performed in conjunction with a Rabbit-α-Ago2 antibody (Abcam, Cambridge, UK, Cat# 186733) and a Mouse-α-LASP1 antibody (Biolegend, San Diego, CA, USA, Cat# 909301). One modification to the recommended protocol by the manufacturer was made during the cell blocking step, where Background Sniper (Biocare Medical, Pacheco, CA, USA, Cat# BS966) was used for 15 min instead of the recommended 1 h with the provided blocking reagent. Bone-Un cells were used in the assay for their larger size compared to the 231S cells, allowing for better visualization of PLA interaction spots. Cells were mounted on coverslips, allowed to attach/spread, and after stimulation were fixed and permeabilized, all as described in previous work [92]. As in the Co-IP assay the cells were serum starved for 1 h prior to stimulation with CXCL12, with a final concentration of 100 nM AMD3465 being added after 30 min of serum starvation for the + CXCL12/+ AMD3465 condition. For the single-antibody control condition the LASP1 antibody was used. After completion of the PLA reaction cells were stained with phalloidin for 10 min at 37 °C (Life Technologies, Carlsbad, CA, USA, Cat# A12379) and DRAQ5 for 5 min at R. T. (Thermo Scientific, Rockford, IL, USA, Cat# 62251) to highlight cell structure and nuclei respectively. Images were taken using the TCS SP5 Laser Scanning Confocal Microscope with corresponding Leica Application Suite X software (Leica, Wetzlar, Germany). All images were processed using the Leica software and quantification of PLA interactions was performed using the ImageJ (Version 1.8.0, https://imagej.nih.gov/) software’s particle analysis function.

### 4.6. Generation of LASP1 Phospho-Mutant Constructs and Phospho-Mutant Cell Lines

All eight LASP1 phosphorylation mutants were generated through site-directed mutagenesis using the QuikChange II kit (Agilent Technologies, Santa Clara, CA, USA, Cat# 200523). Mutagenesis was performed by following the manufacturer’s instructions provided with the QuikChange II kit. Primers for each site are as follows: for Y171F 5′-CAGTGCCCCGGTTTTCCAGCAGCAAAAG-3′ (forward) and 5′-CTGGCGCTGCTGGAAAACCGGGGCACTG-3′ (reverse); for Y171D 5′-CAGTGCCCCGGTTGACCAGCAGCCCCAG-3′ (forward) and 5′-CTGGGGCTGCTGGTCAACC GGGGCACTG-3′ (reverse); for S146A 5′-CAGAGCGTCGGGATGCACAGGACGGCAG-3′ (forward) and 5′-CTGCCGTCCTGTGCATCCCGACGCTCTG-3′ (reverse); for S146D 5′-CCAGAGCGTCGG GATGATCAGGACGGCAGCAGC-3′ (forward) and 5′-GCTGCTGCCGTCCTGATCATCCCGAC GCTCTGG-3′ (reverse). Mutagenesis for each site was performed on the full-length wild-type LASP1 construct in the pGEX-6P-1 vector created/described above. The four single mutant constructs were created first S146A (SA), S146D (SD), Y171F (YF), and Y171D (YD). Phenylalanine replaced alanine as the Y171 phospho-null in order to retain a structure closer to the original Tyrosine. The double mutant constructs S146A/Y171D (SAYD), S146A/Y171F (SAYF), S146D/Y171F (SDYF), and S146D/Y171D (SDYD), were created by using the single mutant constructs as templates and mutating the second phosphorylation site. Verity of the phospho-mutant constructs was confirmed by dideoxy sequencing (Eurofins Genomics, Louisville, KY, USA). The procedure for the LASP1 domain constructs was also utilized here for growing and purifying the mutants. Validation of bead-bound protein was performed by SDS-PAGE followed by Imperial™ Protein Stain. For expression in the Bone-Un LASP1 knockout cell line, these constructs were shuttled from the pGEX-6P-1 vector to the pcDNA3.0 mammalian expression vector (Invitrogen, Carlsbad, CA, USA, Cat# V79020) using the BamHI and XhoI cloning sites. The wild-type LASP1, SAYD, SDYF, and pcDNA3.0 empty vector constructs were electroporated into Bone-Un LASP1 knockout cells. Electroporation was performed using the Cell Line Nucleofector™ Kit V (Lonza Group AG, Basel, Switzerland, Cat# VCA-1003) and the Nucleofector™ 2B device (Lonza Group AG, Basel, Switzerland, Cat# AAB-1001), following the manufacturers recommended instructions for electroporation into the MDA-MB-231 (MDA-MB-231: ATCC^®^ HTB-26™, Manassas, VA, USA) cell line. Selection of WT or mutant-expressing cell lines was done using G418 Sulfate (Gibco™ by ThermoFisher, Waltham, MA, USA, Cat# 11811031) at a concentration of 0.7 mg/mL. Expression of the rescue and mutants was performed by harvesting cell lysates, with analysis done by 10% SDS-PAGE and Western blotting.

### 4.7. Bioinformatic Analysis

Analysis of Let-7a and miR-100 miRNA target expression in patient tissue samples was performed using the online and publicly available Oncomine™ software (https://www.oncomine.org/). Analysis was performed using the settings of breast cancer vs. normal analysis. The two datasets used in the analysis were the TCGA Breast [69] and Curtis Breast [70]. Box and whisker plots of the log2 median centered ratio for the invasive lobular and invasive ductal carcinoma subtypes were created using GraphPad Prism 8 software [93,94].

### 4.8. Luciferase Reporter Assays for miRNA Targets

All luciferase constructs used in this body of work were generously provided by the Tyson V. Sharp Lab of the Queen Mary University of London. All work in creation and details of these luciferase constructs is described in previous work [58]. All luciferase constructs were in the psiCHECK™-2 dual luciferase vector (Promega, Madison, WI, USA, Cat# C8011), which contains a Renilla luciferase protein with a modifiable 3′-UTR and an additional control Firefly luciferase under a separate promoter. Five vectors in total were used in the experiment: empty vector psiCHECK™-2; mTOR, which consisted of the full mTOR 3′-UTR; miR-100 Tar, designed with 5 miR-100 target sites; miR-100 Not-Tar, with 5 miR-100 targets sites containing a mis-matched seed sequence; and Let-7a, which was designed with six Let-7a target sites. 100 ng of each vector was transfected into each of our phospho-mutant and rescue cell lines (Wild-type, SAYD, SDYF, and empty vector cell lines) described above. After transfection the cells were incubated overnight at 37 °C. Harvesting of protein lysate and assessment of both Renilla and Firefly luciferase activity was done with the Dual-Luciferase^®^ Reporter Assay System (Promega, Madison, WI, USA, at# E1910) as per the kit’s instructions. All Renilla values were normalized to internal control Firefly expression prior to further normalization against corresponding construct control (Let-7a vs. EV, mTOR vs. miR-100 Not-Tar, and miR-100 Tar vs. miR-100 Not-Tar).

### 4.9. Western Blotting

Primary antibodies used were as follows: Ago2 (Abcam, Cambridge, United Kingdom, Cat# 186733), Ago1 (Cell Signaling Technology, Danvers, MA, USA, Cat# 5053), LASP1 (Biolegend, San Diego, CA, USA, Cat# 909301), Cyclin D1 (Cell Signaling Technology, Danvers, MA, USA, Cat# 2922), β-tubulin (Sigma-Aldrich, St. Louis, MO, USA, Cat# T0198), Vinculin (Bio-Rad, Hercules, CA, USA, Cat# V284), eIF4G2 (Cell Signaling Technology, Danvers, MA, USA, Cat# D88B6), and CCR7 (Abcam, Cambridge, UK, Cat# ab32527). The two secondary antibodies used in these western blots were Goat anti-Mouse IgG (H + L) Superclonal™ Secondary Ab conjugated to HRP (Thermo Scientific, Rockford, IL, USA, Cat# A28177) or Goat anti-Rabbit IgG (H + L) Superclonal™ Secondary Ab conjugated to HRP (Thermo Scientific, Rockford, IL, USA, Cat# A27036). Development of blots was performed using Amersham™ ECL™ Prime Western Blotting Detection Reagent (GE Healthcare Life Sciences, Pittsburgh, PA, USA, Cat# RPN2232). Detection of developed blots was done by either HyBlot ES™ Autoradiography Film (Denville Scientific, Swedesboro, NJ, USA, Cat# E3212) or the G:BOX Chemi XX6/XX9 imaging system (Syngene, Frederick, MD, USA). Densitometry was performed using ImageJ (Version 1.8.0, https://imagej.nih.gov/) and final values represent fold change compared to the experimental control.

### 4.10. Statistical Analysis, Graph Preparation and Figure Design

Statistical analysis and graph preparation were performed using the GraphPad Prism 8 software. Statistical significance was determined by one-way ANOVA for both the luciferase and PLA assays, with *p*-values of < 0.05 as being significant. All figures were finalized in the CorelDRAW Graphics Suite 2020(Corel, Ottawa, ON, Canada).

## 5. Conclusions

The present novel study demonstrates that LASP1 is capable of binding to Ago2 in a CXCL12-dependent manner, establishing Ago2 as a member of the CXCR4 signaling pathway. We found this interaction occurs in TNBC cells and is driven by the phosphorylation status of LASP1 in response to CXCR4 activity. Furthermore, we revealed that the interaction between these two proteins altered the ability of Ago2 to repress pro-metastasis related Let-7a targets *in vitro*. These results suggest that the role of Ago2 as a master regulator of protein translation can be manipulated in response to CXCR4 activity, leading to a change in cancer progression related targets.

## Figures and Tables

**Figure 1 cancers-12-02455-f001:**
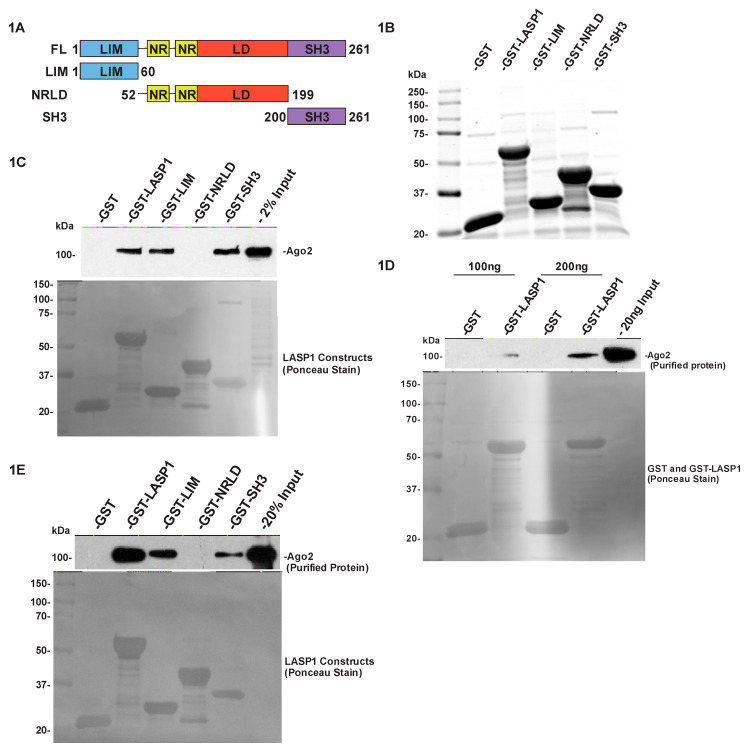
LASP1 directly binds to Ago2. (**A**) Schematic diagram of GST-LASP1 and its domains; FL: Full length LASP1 including all residues 1–261; LIM: LIM domain including residues 1–60; NRLD: Nebulin repeats and linker domain including residues 51–199; SH3: SH3 domain including residues 200–261. (**B**) Imperial Blue Protein Stain of 1.5 nmol of purified GST, GST-LASP1 and its domains separated by 10% SDS-PAGE. (**C**) GST-pulldown of Ago2: 1 mg of 231S lysate was incubated with 1.5 nmol of each GST-tagged protein. Association of endogenous Ago2 was detected by Western blotting and presence of proteins validated by Ponceau S Stain (*n* = 3). (**D**) Direct binding of Ago2 to LASP1: 100 ng or 200 ng of purified recombinant Ago2 was incubated with 1.5 nmol of GST or GST-LASP1. Direct binding of recombinant Ago2 was detected via Western blotting and loading of proteins validated by Ponceau S Stain (*n* = 1). (**E**) Direct binding of Ago2 to LASP1 domains: 100 ng of purified recombinant Ago2 was incubated with 1.5 nmol of each GST-tagged protein. Direct binding of recombinant Ago2 to different domains of LASP1 was detected by Western blotting and loading of proteins was validated by Ponceau S Stain (*n* = 3). Uncropped version of the Western Blots is present in Appendix A.

**Figure 2 cancers-12-02455-f002:**
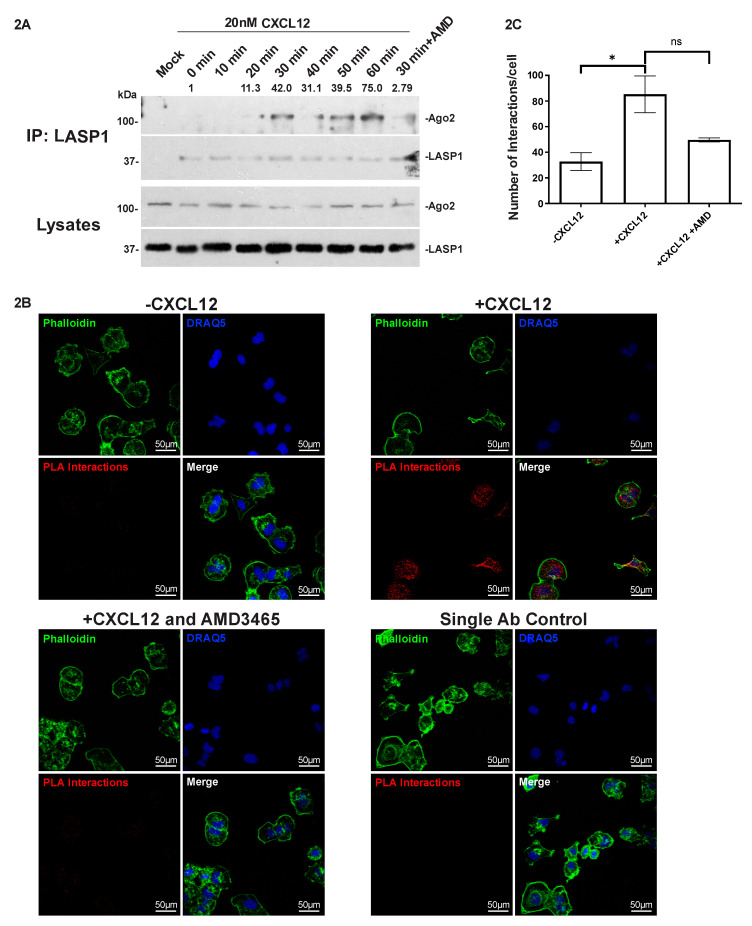
CXCR4-Dependent Association of LASP1 with Ago2 in situ. (**A**) Co-immunoprecipitation of Ago2 with LASP1 after stimulation of 231S cells with 20 nM CXCL12 for 0–60 min, with added unstimulated mock control and 30 min stimulation with added 100 nM AMD3465. Total protein lysates were resolved by 10% SDS-PAGE and immunoblotted for LASP1 and Ago2 (*n* = 3). (**B**) Proximity ligation assay of the LASP1-Ago2 interaction in Bone-Un cells under four conditions: 20 nM CXCL12 stimulation for 30 min(+ CXCL12); 20 nM CXCL12 stimulation for 30 min with pre-treatment of cells with 100 nM AMD3465 for 30 min (+ CXCL12 and AMD3465); unstimulated Bone-Un cells (− CXCL12); and single antibody control with no Ago2 antibody. The interactions were visualized as immunofluorescent dots (pseudo-colored red), the cellular outlines were marker by F-actin staining with Phalloidin (pseudo-colored green), and nuclei by DRAQ5 (pseudo-colored blue). Merged images of all three channels are provided (*n* = 2). (**C**) Quantification of PLA interaction spots by ImageJ (Version 1.8.0, https://imagej.nih.gov/) particle analysis. At least 50 cells from each condition were analyzed for total number of interactions from each PLA experiment after thresholding the particle size. The results are reported as number of interactions per cell with error bars representing SEM. Statistical analysis was performed by one-way ANOVA between − CXCL12 and + CXCL12 (*p* = 0.0127), and between + CXCL12 and + CXCL12 with AMD3465 (*p* = 0.2833). * represents statistical significance (*p* < 0.05), and ns represents no statistical significance (*p* > 0.05). Scale Bar = 50μm. Uncropped version of the Western Blots is available in Appendix A.

**Figure 3 cancers-12-02455-f003:**
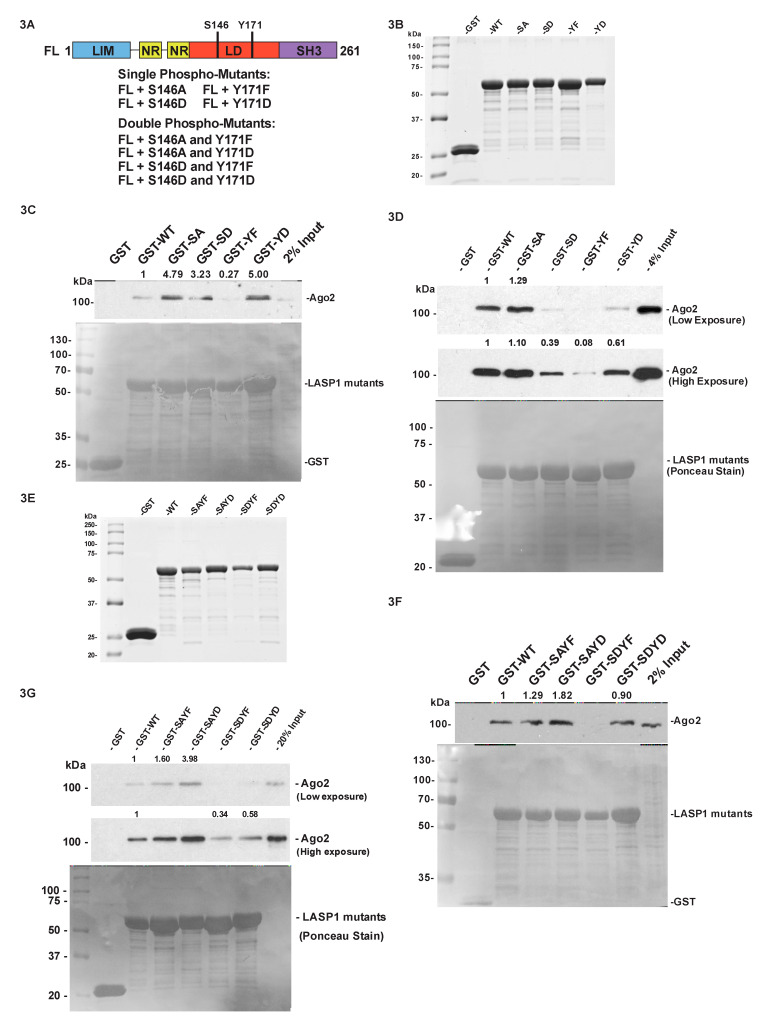
Phospho-Mutants of LASP1 Differentially Interact with Ago2. (**A**) Schematic diagram of designed LASP1 phosphorylation mutants. (**B**) Representative profile of purified GST-LASP1 single phospho-mutants: 1.5 nmol of each purified mutant was resolved by 10% SDS-PAGE and visualized by staining with Imperial Blue stain. (**C**) LASP1 single-mutant pulldown for Ago2: 1 mg of Bone-Un LASP1 KO total lysate was incubated with 1.5 nmol of each LASP1 single-mutant protein. Associated Ago2 was detected by immunoblot analysis and loading of GST-fusion proteins was visualized by Ponceau S staining. (**D**) Direct binding of LASP1 single-mutants to recombinant Ago2: 100 ng of purified recombinant Ago2 was incubated with 1.5 nmol of each purified single-mutant protein. Directly bound Ago2 was detected by immunoblot analysis and loading of GST-fusion proteins was visualized by Ponceau S staining. (**E**) Representative profile of purified GST-LASP1 double phospho-mutants: 1.5 nmol of each of the purified proteins were resolved by 10% SDS-PAGE and proteins were visualized by staining with Imperial Blue stain. (**F**) LASP1 double-mutant pulldown for Ago2: 1 mg of Bone-Un LASP1-KO total lysate was incubated with 1.5 nmol of each of the double-mutants. Associated Ago2 was detected by immunoblot analysis and loading of GST-fusion proteins was visualized by Ponceau S staining. (**G**) Direct binding of LASP1 double-mutants to recombinant Ago2: 100 ng of purified recombinant Ago2 was incubated with 1.5 nmol of purified LASP1 double mutants. Directly bound Ago2 was detected by immunoblot analysis and loading of GST-fusion proteins was visualized by Ponceau S staining. Uncropped version of the Western Blots is available in Appendix A.

**Figure 4 cancers-12-02455-f004:**
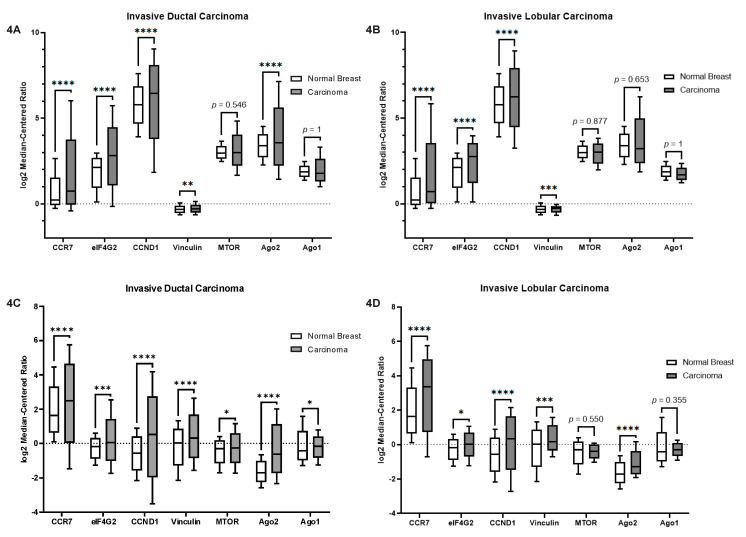
Let-7a and miR-100 targets are upregulated in Breast Carcinomas. Gene expression data from breast carcinoma patients was analyzed using Oncomine™ (https://www.oncomine.org/). Both the Curtis Breast and TCGA Breast datasets were analyzed and results presented as box and whisker plots of the log2 median-centered ratio (Fold change). (**A**) Curtis Breast Invasive Ductal Carcinoma (*n* = 1556) vs. normal breast (*n* = 144). (**B**) Curtis Breast Invasive Lobular Carcinoma (*n* = 148) vs. normal breast (*n* = 144). (**C**) TCGA Breast Invasive Ductal Carcinoma (*n* = 389) vs. normal breast (*n* = 61). (**D**) TCGA Breast Invasive Lobular Carcinoma (*n* = 36) vs. normal breast (*n* = 61). Box edges represent the 25th and 75th percentiles, and whiskers represent the minimum and maximum. ****, ***, **, and * indicate *p* < 0.0001, *p* < 0.001, *p* < 0.01, and *p* < 0.05 respectively, evaluated by Student *t*-test.

**Figure 5 cancers-12-02455-f005:**
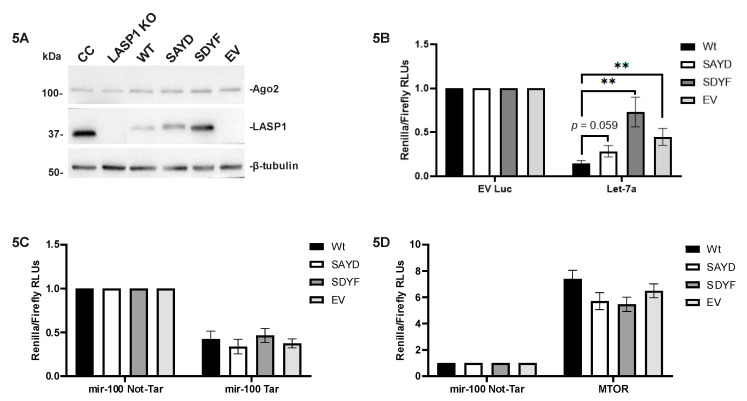
LASP1 Alters Let-7a Guided Ago2 Activity. (**A**) Stable ectopic expression of LASP1 WT and its mutants SAYD and SDYF in the LASP1-KO background. Total lysates from Bone-Un CRISPR control (CC) and LASP1-KO (LASP1 KO) as well as the LASP1-KO cells with expression of re-introduced LASP1 WT and its doublemutants: pcDNA 3.0 empty vector (EV) control, LASP1 wild-type rescue (WT), dominant active phospho-mutant LASP1 (SAYD), and dominant-negative phospho-mutant LASP1 (SDYF). 20 µg of total was resolved by 10% SDS-PAGE and immunoblotted for LASP1. (**B**–**D**) Luciferase assays in LASP1 rescue cell lines with dual-luciferase reporters. LASP1 rescue cell lines WT, SAYD, SDYF, and EV were incubated overnight with 4 µg of each psiCHECK2 dual-luciferase vector containing an internal firefly luciferase control and experimental Renilla luciferase including no 3′-UTR (EV Luc), artificial 3′-UTR containing Let-7a target sites (Let-7a), artificial 3′-UTR containing miR-100 target sites (miR-100 Tar), artificial 3′-UTR containing miR-100 target sites with mis-matched seed sequences (miR-100 Not-Tar), and the inserted 3′-UTR of the mTOR mRNA (mTOR). Cell lysates were harvested from each condition and relative luciferase units (RLUs) measured by luminometer. Final readings for each condition was expressed as the ratio of Renilla to firefly RLUs with Let-7a compared against EV Luc (**B**), miR-100 Tar vs. miR-100 Not-Tar (**C**), and mTOR vs. miR-100 Not-Tar (**D**). All experiments performed *n* = 3 with eight technical replicates per experiment. Uncropped version of the Western Blots is available in Appendix A. ** *p* < 0.01.

**Figure 6 cancers-12-02455-f006:**
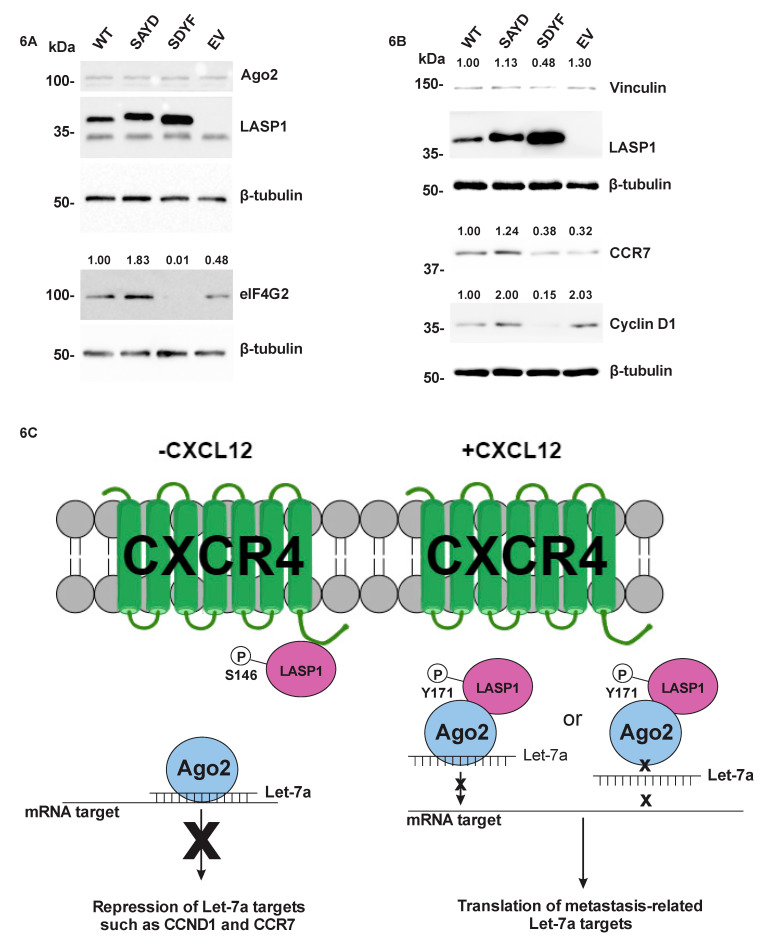
Protein Expression of Let-7a Target Genes are Altered in a LASP1-Ago2 Interaction-Dependent Manner. (**A**) Western blotting of Bone-Un LASP1 KO cell lines with re-introduced LASP1 and its mutants for Ago2 expression and eIF4G2 (*n* = 3). (**B**) Western blotting of additional Let-7a targets vinculin (*n* = 3), CCR7 (*n* = 3), and cyclin D1 (*n* = 3). (**C**) Model of the CXCR4-LASP1-Ago2 pathway in the presence or absence of CXCL12. In the absence of CXCL12, LASP1 is phosphorylated on S146 and remains bound to CXCR4, while Ago2 can freely use Let-7a to repress motility-related targets. In the presence of CXCL12, CXCR4 is stimulated which induces dephosphorylation of LASP1 on S146 and phosphorylation on the Y171 site. Once phosphorylated on Y171 LASP1 can bind to Ago2 and alter Let-7a driven RNAi. Two of the multiple possibilities of how LASP1 could prevent Let-7a driven RNAi are illustrated; first, where LASP1 binds to Let-7a-bound Ago2 and interferes with target binding; second, where LASP1 binds to Ago2 and prevents Ago2 binding to Let-7a. Uncropped version of the Western Blots is available in Appendix A.

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
