# Peer review of "The CXCR4-Dependent LASP1-Ago2 Interaction in Triple-Negative Breast Cancer"

_cancers, 2020, doi:10.3390/cancers12092455_

Round 1

Reviewer 1 Report

The study by Tilley et al from Dr. Raman group at the University of Toledo Health Science Center analyzed the functional interaction between LASP1 (LIM and SH3 protein 1), an actin- binding protein, and Ago2 (Argonaute 2), a protein involved in RNA silencing in triple-negative breast cancer (TNBC) cells. In this study, using the GST-pull-down assays, the authors have convincingly shown that LASP1 directly interact with Ago2 through LIM and SH3 domains but not NRLD domain. Further, this interaction is specific only with Ago2 but not other Argonaute family members. Using co-immunoprecipitation (Co-IP) and proximity ligation assay (PLA), they have also shown the endogenous interaction of LASP1 and Ago2 in TNBC cell line and this interaction is facilitated by CXCL12. Using site-directed mutagenesis assay, they have shown that SAYD double-mutation is strongly associated with Ago2 to LASP1 interaction and SDYF double-mutation is weakly associated with Ago2 to LASP1 interaction. Further, they have identified targets of LASP1-Ago2 interaction such as Let-7a and mir-100 and found that functional LASP-1 reduce Let-7a transcription, by increasing the promoter activity, and dominant negative mutant activate Let-7a transcription. However, LASP-1-Ago2 interaction did not affect mir-100 or mTOR. In addition, they have shown that the stronger interaction of LASP-1-Ago2 (SAYD) promotes the expression of pro-metastatic proteins like Cyclin D1, eIF4G2, CCR4, and Vinculin. Based on these finding, they have developed a working model in which in the presence of CXCL12, phosphorylated LASP1 interact with Ago2 and this complex may bind to Let-7a, which in turn block the translation of metastasis related genes.

Overall the study has well developed and executed. Results are an outstanding quality. Experimental designs are well articulated and described very well. The results are well discussed and references are well organized.

Overall this is an outstanding manuscript for the readers of Cancers. However, the authors need to clarify the following concerns:

Although the authors have provided valid reason why they selected Let-7a and mir-100 as targets for LASP-1-Ago2, still not convincingly explain why they specifically selected these two targets. They need to discuss more and provide more valid reasons for their justification.

The authors have compared targets of Let-7a and mir-100 in both invasive ductal carcinoma and invasive lobular carcinoma and interpreted with Ago2 but they did not correlate with LASP1. Further, in   Figure 4A. Labels are missing, Normal vs cancer. MTOR label needs to be replaced as "mTOR".

Figure 4B. MTOR label needs to be replaced as "mTOR".

Figure 4C. Labels are missing. MTOR label needs to be replaced as "mTOR".

Figure 4D. MTOR label needs to be replaced as "mTOR".

Reviewer 2 Report

In this study, Tilley and colleagues explored the possibility that LASP1, a key molecule in the CXCR4 signaling pathway in TNBC, interacts with Ago2. They reported that LASP1 interacts with Ago2 which can be stimulated by CXCR4 and this interaction impacts pro-motility and cell cycle proteins. Taken their results together, they proposed a model where CXCR4 signaling activity stimulates binding of LASP1 to Ago2, which promotes cell motility by shifting RNAi-based repression away from pro-motility targets. Although this paper has potential importance to understand the tumorigenesis of TNBC, there are several major points the authors have to answer.

  1. In the author’s previous study, they identified eukaryotic initiation 4F complex (eIF4F) as a novel interaction partner of LASP1 through a proteomic screening and they concluded that CXCR4-LASP1-eIF4F axis would contribute to the preferential translation of oncogenic mRNAs leading to breast cancer progression and metastasis. Therefore, in this LASP1 co-immunoprecipitation analysis, both of eIF4F and Ago2 are expected to be co-precipitated. However, there is no result and discussion about this point. Also, there is no observation about their (eIF4F and Ago2) interaction or role in the CXCR4-LASP1 pathway.
  2. The data set used in in Figure 4 are not TNBC. If the authors intend to support the biological implication of LASP1-Ago2 interaction in TNBC identified by in vitro analysis in this study, the primary cancer dataset must be TNBC.
  3. There is no functional analysis data (migration, invasion etc) after Ago2 knock-out.

Reviewer 3 Report

Minor points

Introduction

Line 35. Indique the reason by which TNBC has high incidence of metastasis , particularly to the lungs and brain as compared with other types of breast cancer.

Line 60-62. ¨S146 is  phosphorylated by protein kinases A and G (PKA and PKG), dephosphorylated by protein  phosphatase 2B (PP2B) [14], and is involved in re-localization of LASP1 from cell membrane  extensions and cell-cell contacts to the cytosol [23]¨. Please, add details on the relevance of phosphorilation/dephosphorilation of these proteins in the context of CXCR4-dependent tumorogenesis (i fis appropiated).

Line 27. ¨CXCR4 is a chemokine receptor involved in several processes during embryogenesis, brain development [27]¨. This study is GREAT, but this cyte reflects more neuroinflammatory roles that neurodevelopment roles of CXCR4¨. I suggest replace this reference by another more more appropiate about CXCR4 and brain development.

The same for cyte 28. There are studies which evaluate the role of CXCR4 in neurons and  stem cells migration. For example,  these references:

José Joaquín Merino, Victor Bellver-Landete, María Jesús Oset-Gasque, Beatriz Cubelos CXCR4/CXCR7 molecular involvement in neuronal and neural progenitor migration: focus in CNS repair. J Cell Physiol 2015 Jan;230(1):27-42.  doi: 10.1002/jcp.24695.

Shimizu Saori, Brown Michael,  Sengupta Rajarshi,  Penfold Mark E,  Meucci Olimpia. CXCR7 protein expression in human adult brain and differentiated neurons. PLoS One. 2011;6(5):e20680. doi: 10.1371/journal.pone.0020680. Epub 2011 May 31.

Which LASP1 dependent signaling pathways are involved in downstream signaling pathways for promoting invasión?

¨We also found that this interaction impacts pro-103 motility and cell cycle proteins that are targets of Ago2 based RNAi regulation including the 104 chemokine receptor CCR7, cyclin D1 and eukaryotic initiation factor G2 (eIF4G2)¨. Please, describe the conexión between CCR7, CXCR4 and these proteins in the context of cell cycle for tumoral progression.

Results

In my opinion, the normalization with Pouncea red is not correct as loading control of proteins in western blot. We should use beta-actine (41 Kda similar to chemokines) or 3PI kinasa subumits (p110, p85), beta-actinine but not red Pouncea. In fact, GST-SH3 line is difficult to see here by Pounceau red staining

Thus, the results in figure 1 C are not conclusive for me. Please, provide a load control marker such as beta-actinine. I undestand that beta active is closet o molecular weight of chemokines but it also better to do stripping than use Pouncea red as loaded control for proteins in western blot.

Line 119. ¨ Immunoblotting of the pulldown 119 revealed that endogenous Ago2 associates with full length LASP1 as well as its LIM and SH3 domains 120 (Figure 1C)¨. However, loading protein controls are not the best because Paunceau read should be changed by beta-actine or another loading control (with different molecular weigth). In fact, the line of western for GST-SH3 must be imagined in the blot.

Line 123. Please, indicate the reason by which Ago2 is the only family member  with catalytic activity.

Line152. Please, explain the reason by which we have not used AMD-3100. Then, explain why we use AMD365 since both are CXCR4 chemokine blockers. Is there particular reasons?

Line 154. ¨ We found that the addition of AMD3465 prior to stimulation with CXCL12 ablated the peak observed at 30 min. This suggested that the interaction seen at 30 min was specific to CXCR4¨. Yes, but you have not tests at 60 min (20 nM CXCL12). This is strange for this reviewer because I see twice as compare with 30 min data (20 nM CXCL12) exposure for LASP-1 protein in coinmunoprecipation studies from figure 2A. Please, can explain these differences and add this information to result section (fig 2a). The inmunofluorescence analysis is very clear and clearly support their findings (fig 2 C).

Line 161-168. Please, explain how exactly evaluate the increase in both the number and intensity of PLA interactions (dots) when stimulated with CXCL12 (Figure 2B)? Have you measured fluorescence intensity  by Metaphorm? Which image software you use for this quantification as well as whihc statistal analysis we chose for quantification of PLA number interations ?

Line 272. Please, explain better the use of in the Curtis Breast Invasive Ductal Carcinoma (IDC) dataset (Figure 4A) as well as  both of the “The Cancer Genome Atlas” in the present study.

The figure 4A and 4B showed a big variability for CCR7. Is there a significant effect (p<0.05) vs normal breast cells here? Please,  verify it.

Line 306. ¨both the EV cell line and the dominant-negative SDYF cell lines both increased renilla luciferase expression over the WT rescue cell line¨. Please, explain with more this detail this contradictory result.

Line 336. These authors used a correct loading control here (β-tubulin loading).

Why LASP1 has two bands in figure 6a and only one band in figure 6b? This is strange for this reviewer. I would recommend you to include beta-tubulin ad the end of western also.

Which is the relationship between CXCR4-Argo2 and CCR7 in the present study if CXCL2 does not bind to CCR7?

The discussion is brillant and contains all described results, which has been very well discussed and relationed with published studied in the field.

This is brillant study in general but

In general, please explain why you have chosen these genes, incluging eIF4G2 (Figure 6A), vinculin, CCR7, 333 and cyclin D1 (Figure 6B) but not others in the present study.

Round 2

Reviewer 2 Report

I have no further comment.